# Ancient European dog genomes reveal continuity since the Early Neolithic

Laura R. Botigué[1,*], Shiya Song[2,*], Amelie Scheu[3,4,*], Shyamalika Gopalan[1], Amanda L. Pendleton[5], Matthew Oetjens[5], Angela M. Taravella[5], Timo Seregély[6], Andrea Zeeb-Lanz[7], Rose-Marie Arbogast[8], Dean Bobo[1], Kevin Daly[4], Martina Unterländer[3], Joachim Burger[3], Jeffrey M. Kidd[2,5] & Krishna R. Veeramah[1]

Europe has played a major role in dog evolution, harbouring the oldest uncontested Palaeolithic remains and having been the centre of modern dog breed creation. Here we sequence the genomes of an Early and End Neolithic dog from Germany, including a sample associated with an early European farming community. Both dogs demonstrate continuity with each other and predominantly share ancestry with modern European dogs, contradicting a previously suggested Late Neolithic population replacement. We find no genetic evidence to support the recent hypothesis proposing dual origins of dog domestication. By calibrating the mutation rate using our oldest dog, we narrow the timing of dog domestication to 20,000–40,000 years ago. Interestingly, we do not observe the extreme copy number expansion of the *AMY2B* gene characteristic of modern dogs that has previously been proposed as an adaptation to a starch-rich diet driven by the widespread adoption of agriculture in the Neolithic.

[1] Department of Ecology and Evolution, Stony Brook University, Stony Brook, New York 11794-5245, USA. [2] Department of Computational Medicine and Bioinformatics, University of Michigan, Ann Arbor, Michigan 48109, USA. [3] Palaeogenetics Group, Johannes Gutenberg-University Mainz, 55099 Mainz, Germany. [4] Smurfit Institute of Genetics, Trinity College Dublin, Dublin 2, Ireland. [5] Department of Human Genetics, University of Michigan, Ann Arbor, Michigan 48109, USA. [6] Department of Prehistoric Archaeology, Institute of Archaeology, Heritage Sciences and Art History, University of Bamberg, 96045 Bamberg, Germany. [7] Generaldirektion Kulturelles Erbe Rheinland-Pfalz, Direktion Landesarchäologie, Außenstelle Speyer, 67346 Speyer, Germany. [8] CNRS UMR 7044-UDS, 5 Allée du Général Rouvillois F 67083 Strasbourg, France. * These authors contributed equally to this work. Correspondence and requests for materials should be addressed to K.R.V. (email: krishna.veeramah@stonybrook.edu).

Europe has been a critically important region in the history and evolution of dogs, with most modern breeds sharing predominantly European ancestry[1]. Furthermore, the oldest remains that can be unequivocally attributed to domestic dogs (*Canis lupus familiaris*) are found on this continent, including an Upper Palaeolithic 14,700-year-old jaw-bone from the Bonn–Oberkassel site in Germany[2] (older specimens from Siberia and the Near East that have been proposed remain highly controversial[3,4]). While ancient mitochondrial DNA (mtDNA) suggests a European centre of dog domestication[5], analyses of mitochondrial and genomic data from modern dogs have suggested East Asia[6,7], the Middle East[8] and Central Asia[9].

The Neolithic period in Central Europe ranges from ~7,500 to 4,000 BP and can be further subdivided based on specific features of human culture[10] (Supplementary Table 1). Multiple studies have found evidence of a prehistoric turnover of canid mtDNA lineages sometime between the Late Neolithic and today, with haplogroup C, which appears in almost all Neolithic dogs but in less than 10% of modern dogs, being replaced by haplogroup A in most of Europe[5,11,12]. By analyzing genomic data from modern dogs and a Late Neolithic (~5,000 years old) Irish dog from Newgrange (hereafter referred to as NGD), Frantz *et al.*[12] argue that this matrilineal turnover was a consequence of a major population replacement during the Neolithic. However, NGD primarily shares ancestry with modern European dogs, implying the proposed population replacement had largely already occurred before this individual lived. Frantz *et al.*[12] also estimate a relatively recent east–west dog divergence (14,000–6,000 years ago), which, placed within the context of existing archaeological data, they explain with a dual origin of dog domestication.

The characterization of samples from earlier in the Neolithic and from continental Europe is necessary to examine whether and to what extent a large-scale demographic replacement occurred during this period. This would be evidenced by a distinct ancestry absent in modern dog genomes that was more prominent in dogs from earlier in the Neolithic, as opposed to genomic continuity from the Early Neolithic to today. Therefore, we present analysis of ~9× coverage whole genomes of two dog samples from Germany dating to the Early and End Neolithic (~7,000 years old and ~4,700 years old, respectively). We observe genetic continuity throughout this era and into the present, with our ancient dogs sharing substantial ancestry with modern European dogs. We find no evidence of a major population replacement; instead, our results are consistent with a scenario where modern European dogs emerged from a structured Neolithic population. Furthermore, we detect an additional ancestry component in the End Neolithic sample, consistent with admixture from a population of dogs located further east that may have migrated concomitant with steppe people associated with Late Neolithic and Early Bronze age cultures, such as the Yamnaya and Corded Ware culture[13]. We also show that most autosomal haplotypes associated with domestication were already established in our Neolithic dogs, but that adaptation to a starch-rich diet likely occurred later. Finally, we obtain divergence estimates between Eastern and Western dogs of 17,000–24,000 years ago, consistent with a single geographic origin for domestication, the timing of which we narrow down to between ~20,000 and 40,000 years ago.

## Results

### Archaeological samples and ancient DNA sequencing.
The older specimen, which we refer to hereafter as HXH, was found at the Early Neolithic site of Herxheim and is dated to 5,223–5,040 cal. BCE (~7,000 years old) (Supplementary Fig. 1). The younger specimen, which we refer to hereafter as CTC, was

found in Cherry Tree Cave and is dated to 2,900–2,632 cal. BCE (~4,700 years old), which corresponds to the End Neolithic period in Central Europe[14] (Supplementary Fig. 2 and Supplementary Notes 1–3).

We generated whole-genome sequence data for the two ancient dogs and mapped over 67% of the reads to the dog reference genome (CanFam3.1), confirming high endogenous canine DNA content for both samples (Supplementary Table 2 and Supplementary Note 4). MapDamage[15] analysis demonstrated that both samples possess damage characteristics typical of ancient DNA[16] (Supplementary Fig. 3). The final mean coverage for both samples was ~9×, while coverage on the X and Y chromosome was ~5×, indicating they both are male. We also reprocessed the NGD data[12] using the same pipeline as for CTC and HXH. To call variants, we used a custom genotype caller implemented in Python (see Supplementary Note 5) that accounts for DNA damage patterns[17]. We found that our approach eliminated many false positives that are likely due to postmortem damage (Supplementary Fig. 4).

**Modern canid reference data sets.** We analysed these Neolithic dogs within the context of a comprehensive collection of 5,649 canids, including breed dogs, village dogs and wolves that had been previously genotyped at 128,743 single nucleotide polymorphisms (SNPs)[9,18] (Supplementary Table 3), as well as 99 canid whole genomes sequenced at medium to high coverage (6–45×) (Supplementary Table 4). To account for biases in variant calling that might occur as a result of this variable coverage, we ascertained variable sites in an outgroup (such that mutations are known to have occurred in the root of all the populations being analysed). We explored different ascertainment schemes for the whole-genome data (Supplementary Note 6) and chose to use a call set that includes sites variable in New World wolves (we note, though, that our primary results are robust to changes in the ascertainment scheme). This call set contains 1,815,911 variants that are likely either private to New World wolves or arose in the grey wolf (*Canis lupus*) ancestral population, and thus is the least biased with regard to their ascertainment in Old World wolves and dogs.

**mtDNA analysis.** We examined the phylogenetic relationship of the entire mitochondrial genomes of HXH and CTC with a comprehensive panel of modern dogs across four major clades (A–D), modern wolves and coyotes, and previously reported ancient wolf-like and dog-like whole mitochondrial sequences[5,12]. Like other European Neolithic dogs, both HXH and CTC belong to haplogroup C (Fig. 1a, Supplementary Fig. 5 and Supplementary Note 7) together with NGD and the Upper Palaeolithic 12,500-year-old Kartstein Cave dog (also from Germany). We note that Bonn–Oberkassel also falls in the same haplogroup[5] (Supplementary Figs 6 and 7), although analysis of this sample is complicated by low mtDNA sequence coverage, pointing to some degree of matrilineal continuity in Europe over ~10,000 years, ranging from the Late Palaeolithic to almost the entire Neolithic. The inclusion of 24 additional clade C samples[19] in the phylogenetic analysis reveals the expected C1 and C2 split (100% support) and that HXH, CTC, NGD and the Kartstein Cave dog share a common lineage with C1 dogs (Supplementary Fig. 8). This topology suggests that these ancient European dogs belong to an older sub-haplogroup that is sister to the progenitor of the C1b and C1a sub-haplogroups and possibly absent in modern dog populations.

**Genomic clustering of the European Neolithic dogs.** We constructed a neighbour-joining (NJ) tree using the whole-genome sequence data set (Fig. 1b and Supplementary Figs 9 and 10) to

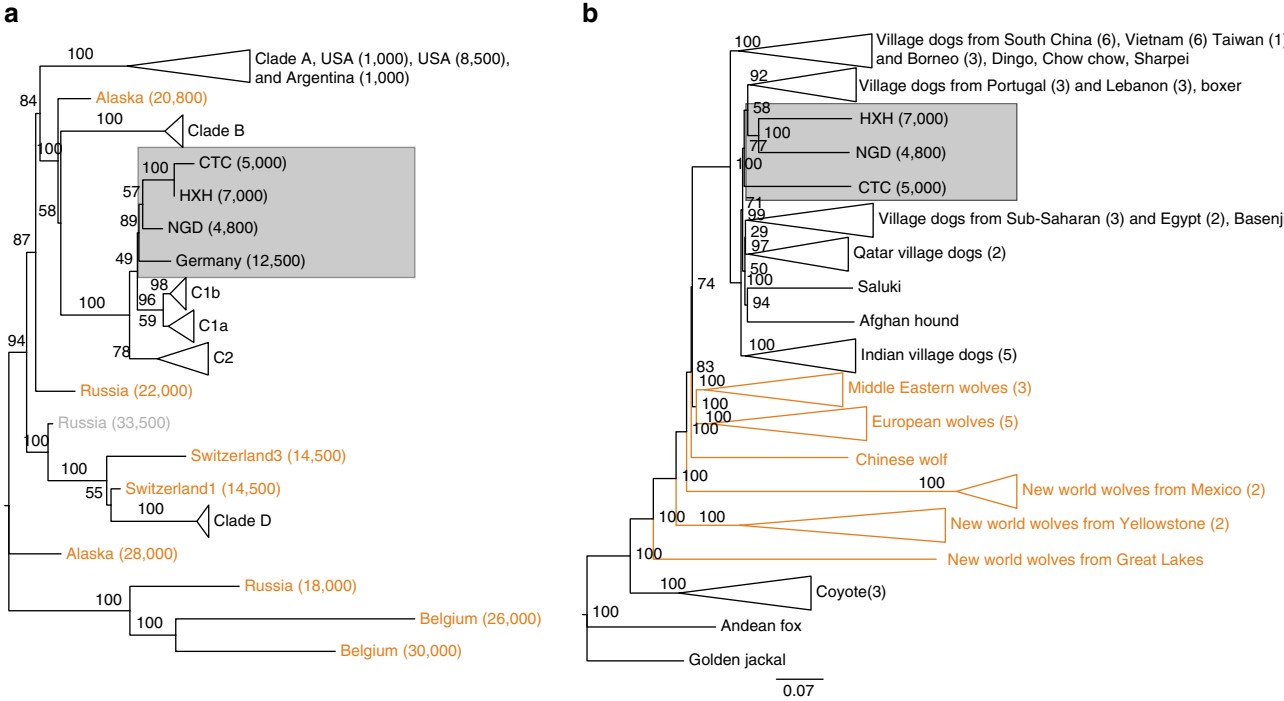

**Figure 1 | Phylogeny of ancient and contemporary canids. (a)** Phylogeny based on mtDNA. Age of the samples is indicated in parentheses, wolf samples are shown in orange. **(b)** NJ tree based on pairwise sequence divergence from whole-genome data.

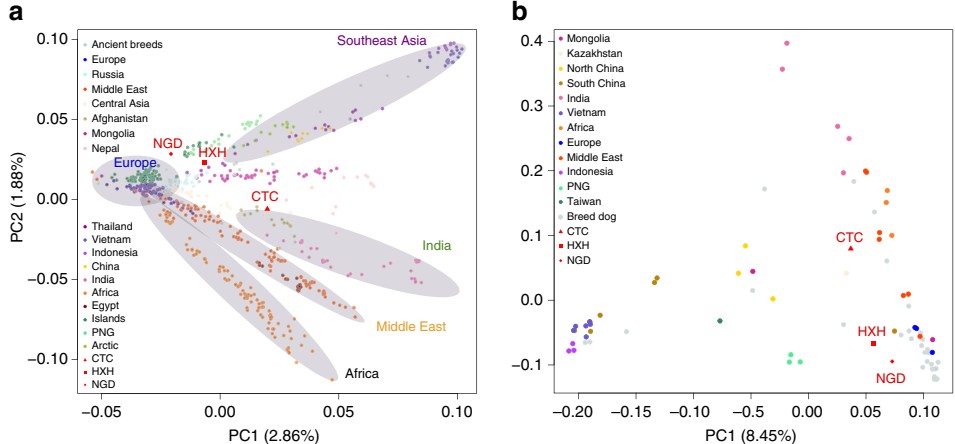

**Figure 2 | PCA between ancient and contemporary canids. (a)** PCA of village dogs, with breed dogs and ancient dogs projected onto the PC space using SNP array data. **(b)** PCA of village dogs, breed dogs and ancient dogs using whole-genome SNP data ascertained in the New World wolves.

determine which modern dog population shows the greatest genetic similarity to the ancient samples (Supplementary Note 8). We found that the Early Neolithic HXH and Late Neolithic NGD grouped together as a sister clade to modern European village dogs, while CTC was external to this clade, but still more similar to it than to any other modern population. As shown previously[12], East Asian village dogs and breeds are basal to all other dogs.

We also performed a principal component analysis (PCA) using both the SNP array and whole-genome data (Fig. 2a,b and Supplementary Note 9), with both data sets showing highly similar population structure patterns, despite having very different ascertainment schemes (SNP array data are expected to be biased towards European breed dogs). The larger SNP array reference data set shows that village dogs primarily separate into five distinct geographic clusters: Southeast Asia, India,

Middle East, Europe and Africa. Breed dogs fall mostly within European village dogs' variation with the exception of basal or 'ancient' breeds[20]. Consistent with NJ tree analysis, all three ancient samples fell within the range of modern dog variation. HXH and NGD are the ancient samples found closest to the major European cluster, both lying adjacent to the cluster of Pacific Island dogs that are thought to be derived almost completely from European dogs[9]. CTC is located next to village dogs from Afghanistan, a known admixed population also inferred to have a major European-like ancestry component, as well as potential contributions from South and East Asian populations[9].

We note that the position of NGD in our reanalysis does not agree with that reported in Frantz et al.[12], where it lies as an outlier in PC2. Such deviation was interpreted by Frantz et al.[12]

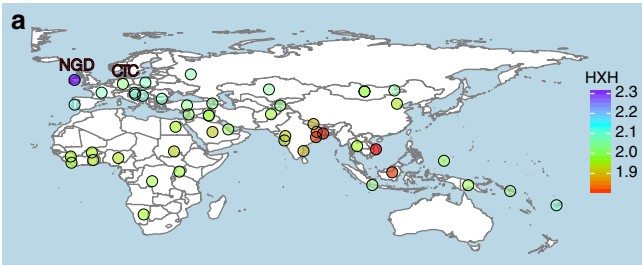

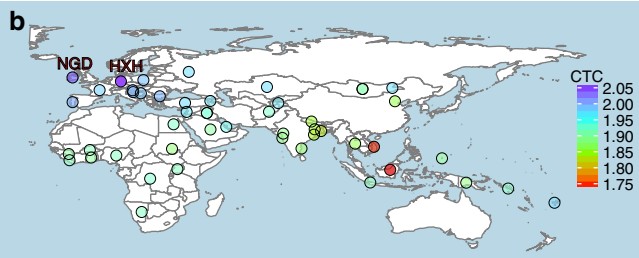

**Figure 3 | Genetic affinity of ancient samples.** Heat map of outgroup-*f3* statistics of the form *f3* (golden jackal; ancient sample; X) based on SNP array genotype data. Higher *f3* values indicate increased shared drift between the samples, and therefore higher genetic similarities. (**a**) HXH shows greatest similarity with NGD and modern European village dogs (higher *f3* values), and is most distant to East Asian and Indian village dogs. (**b**) CTC shares the most genetic similarity with HXH, followed by NGD and other European dogs. In addition, CTC shows greater similarity to village dogs from India (particularly unadmixed populations in the east) than HXH does. Outgroup-*f3* statistic maps were created using the R packages *ggplot2* and maps using the public domain Natural Earth data set.

as NGD carrying ancestry from an extinct European population. However, we found that this is due to a technical artifact that occurred because of the inclusion of both the uncalibrated and calibrated version of this ancient genome in the same PCA. Once one of the duplicate data points is removed from the sample set, NGD returns to the modern dog cluster (Supplementary Figs 11 and 12). Fundamental differences in the overall distribution of genetic variance in PC space between the two studies are due to the overall ascertainment of samples. Since our data set contains a substantially more diverse collection of samples, we assert that our PCA results are more reflective of the true range of dog diversity.

We further examined the genetic relatedness between ancient and modern dogs by performing an *f3*-outgroup analysis[21,22] on both the SNP array and whole-genome sequence data sets. We used the golden jackal and Andean fox as outgroups for the SNP array and the whole-genome data sets, respectively. Our results corroborated NJ tree and PCA findings, and showed that all three Neolithic European samples are genetically most similar to modern European dogs (Fig. 3, Supplementary Figs 13 and 14 and Supplementary Note 10).

**Evidence of admixture in Neolithic dogs.** Our results are consistent with continuity of a European-like genetic ancestry from modern dogs through the entire Neolithic period. However, the slightly displaced position of the ancient samples from the European cluster in the PCAs (particularly for CTC) suggests a complex history. We therefore performed unsupervised clustering analyses with ADMIXTURE (SNP array data; Supplementary Fig. 15) and NGSadmix (whole-genome data; Fig. 4 and Supplementary Fig. 16) (Supplementary Note 9) and found that, unlike

contemporary European village dogs, all three ancient genomes possess a significant ancestry component that is present in modern Southeast Asian dogs. This component appears only at very low levels in a minority of modern European village dogs. Furthermore, CTC harbours an additional component that is found predominantly in modern Indian village as well as in Central Asian (Afghan, Mongolian and Nepalese), and Middle Eastern (Saudi Arabian and Qatari) dogs (concordant with its position in the PCA), as well as some wolf admixture.

We formally modelled these potential admixture events by applying the tree-based framework MixMapper[23] to both the SNP array and whole-genome data (Supplementary Note 10). This approach interrogates every pair of branches in a scaffold tree to infer putative sources of admixture for target samples (in this case HXH, CTC and NGD) via the fitting of *f*-statistics. We constructed the scaffold trees (Supplementary Figs 17 and 18) excluding those populations that showed evidence of admixture as determined by an *f3* statistic test (Supplementary Tables 5 and 6). MixMapper inferred that HXH and NGD were both formed by an admixture event involving the ancestors of modern European and Southeast Asian dogs (Supplementary Tables 7–9), with ~19–30% gene flow from the latter into HXH and NGD, as estimated by an *f4*-ratio test (Supplementary Table 10). Analysis with ADMIXTUREGRAPH[22] on the whole-genome data, a method related to MixMapper that examines a manually defined demographic history, demonstrated a perfect fit for the observed *f*-statistics under this model (Supplementary Fig. 19 and Supplementary Note 11).

To disentangle the more complex admixture patterns observed in CTC, we first sought to understand its relationship to HXH given that both samples originate from Germany. Our *f3*-outgroup analysis revealed that CTC had greater affinity with HXH than with any modern canid or with NGD (Fig. 3b, Supplementary Figs 13b and 14b and Supplementary Note 10). We therefore performed a MixMapper analysis where HXH was set as one of the sources of admixture for CTC, which identified a population ancestral to modern Indian or Saudi Arabian village dogs as the second source of admixture (Supplementary Table 11). Further support for genetic continuity between HXH and CTC was found using ADMIXTUREGRAPH when two alternative demographic models were tested. In the first model (model A), CTC descends from the same population as HXH followed by admixture with an Indian-like population, while in model B both ancient samples descend from independently diverged European lineages (and therefore there was no genetic continuity between the two). Model A provides a much better fit to the data (Fig. 5a and Supplementary Fig. 20), producing only two *f4* outliers (no *f2* or *f3* outliers), one of which was barely significant ($Z = 3.013$). Model B produced 74 outliers (Supplementary Note 11). Even though there is the risk of overfitting the model to the data, the stark difference between the two models points to continuity among German dogs during the Neolithic, along with gene flow into CTC at the end of this era from an outside source carrying the genetic component observed in contemporary Middle Eastern and Central and South Asian dog populations.

We investigated the wolf admixture inferred by the unsupervised clustering analysis with SpaceMix[24], a method that creates a baseline 'geogenetic' map that can be used to detect deviations in patterns of covariance that may reflect long-distance admixture (Supplementary Note 9). The clustering of modern and ancient dogs in SpaceMix is essentially the same as observed in the PCA (Supplementary Fig. 21). However, it additionally inferred around 10% ancestry in CTC (but not HXH or NGD) from the geogenetic space containing Old World wolves (Supplementary Fig. 22). *f4* statistics of the form *f4*(CTC, HXH, Wolf, Outgroup)

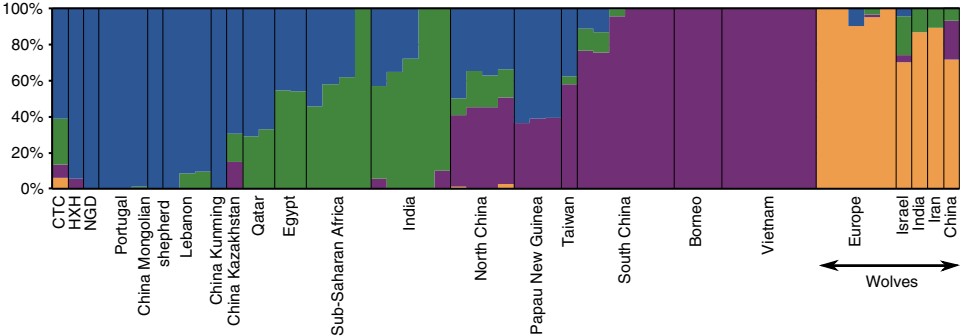

**Figure 4 | Population structure between ancient and contemporary canids.** NGSadmix clustering for $K = 4$ for village dogs, ancient dogs and Old World wolves based on the whole-genome SNP data.

**Figure 5 | Demographic model regarding ancient and contemporary dogs and wolves. (a)** The best model fit to both modern and ancient canid data using ADMIXTUREGRAPH on the whole-genome data set. This model had four $f4$-statistic outliers. Branches are indicated by solid black lines (adjacent numbers indicate estimated drift values in units of $f2$ distance, parts per thousand), whereas admixture is indicated by coloured dashed lines (adjacent numbers indicate ancestry proportions). Sampled individuals/populations are indicated by solid circles with bold outline. Wolves are labelled as 'wolf' and dogs are labelled according to their continental origin. **(b)** Divergence times of contemporary dogs and wolves inferred using G-PhoCS. Mean estimates are indicated by squares with ranges corresponding to 95% Bayesian credible intervals. Migration bands are shown in grey with associated value representing the inferred total migration rates (the probability that a lineage in the target population will migrate into the source population). The divergence time for HXH and NGD and modern European dogs is inferred using a numerical approach. The proportion of Indian village dog ancestry in CTC is inferred by NGSadmix and the proportion of South China village dog ancestry in HXH and NGD is inferred by $f4$-ratio test, shown in red.

suggest that the origin of this wolf component is related to contemporary Iranian/Indian wolves (Supplementary Table 12). Considering that modern Indian dogs show the largest proportion of non-European ancestry detected in CTC, it is possible that an ancestral dog population carrying both this Asian

dog and wolf ancestry admixed with the European population represented by CTC. We find support for this scenario with ADMIXTUREGRAPH, where a model for CTC incorporating modern village dogs and wolves produces no outliers when we allow for an admixture event between European and Indian

village dog lineages along with previous wolf gene flow into the Indian lineage (Supplementary Fig. 23 and Supplementary Note 11).

The complex pattern of admixture found in CTC is similar to that observed in many modern dog populations in Central Asia (such as Afghanistan) and the Middle East, as shown in our unsupervised clustering analyses (Supplementary Figs 15 and 16). This raises the question of whether CTC and these modern dog populations share a common admixture history and are descended from the same ancestral populations. We performed a MixMapper analysis that included HXH in the scaffold tree and observed that the European-like component of CTC is drawn exclusively from this Early Neolithic German dog population (Supplementary Table 13). To the contrary, modern Afghan dogs generally demonstrate inferred ancestry from modern European village dogs. This suggests that modern Afghan village dogs and CTC are the result of independent admixture events.

**Demographic model and divergence time**. The distinct genetic makeup of the European Neolithic dogs compared to modern European dogs indicates that while ancient and contemporary populations share substantial genomic ancestry, some degree of population structure was likely present on the continent. Neolithic dogs would thus represent a now extinct branch that is somewhat diverged from the modern European clade. In addition, our best fit model of modern and ancient canid demography using ADMIXTUREGRAPH involved a topology that would be consistent with a single dog lineage diverging from wolves (Fig. 5a and Supplementary Table 14). Therefore, we attempted to infer the divergence time of HXH and NGD from modern European dogs after the divergence of the Indian lineage that, according to the NJ tree analysis, is the sister clade of the Western Eurasian branch. We note that this is a simplistic bifurcating model of what may have been more complex European geographic structuring and long-term Eurasian dog gene flow.

We first performed a coalescent-based G-PhoCS[25] analysis of the model in Fig. 5b to obtain estimates of divergence time and population diversity (Supplementary Table 15 and Supplementary Note 12). Analysis was performed on sequence data from 16,434 previously identified 1 kb-long loci[26]. Unlike the SNP-based analysis described above, single-sample genotype calling was performed with no particular ascertainment scheme, and we restricted our analysis to eight canid genomes with coverage ranging from 8 to 24×. When we included only modern dogs, we observed that wolf populations appeared to diverge rapidly, concordant with previous studies[26,27], whereas the branching of the main dog lineages took place over a much longer period of time. We found that the (uncalibrated) dog–wolf divergence time in units of expected numbers of mutations per site ($0.5247 \times 10^{-4}$) was similar to that reported in Freedman et al.[26]; however, our dog divergence time ($0.2786 \times 10^{-4}$) was younger than the Freedman et al.[26] estimate, but similar to the Wang et al.[7] estimate, most likely as a result of using Southeast Asian village dogs rather than the dingo (Supplementary Table 16). We also found that the effective population size of village dogs was 5–10-fold higher than that of the boxer.

When we included the ancient samples in the G-PhoCS analysis, all divergence times increased markedly (except the boxer-European village dog split). It is likely that these results are due to remnant postmortem damage artificially inflating variation in the ancient samples and elongating the branch lengths in the G-PhoCS analysis, as we detected an excess of private variants in all three ancient samples compared to European village dogs. We therefore devised a new method for

estimating the HXH/NGD-European split time ($\tau_1$) utilizing G-PhoCS results only for the modern samples and that would be robust to biases resulting from the use of ancient samples (Supplementary Note 13 and Supplementary Fig. 24). Specifically, we calculated the relative observed amount of derived allele sharing exclusive to European village dogs and HXH/NGD versus that exclusive to European and Indian village dogs. The two major advantages of this estimate are that (a) it only depends on previously discovered variable sites in higher coverage modern dogs (our genotype calling in ancient samples is likely to be much more accurate in such situations), and (b) it uses only a single chromosome from each population (which can be randomly picked), and thus does not require calling heterozygotes accurately (that is, it should not be sensitive to the lower coverage of our ancient samples). As expected, European dogs share more derived alleles with the ancient dogs than Indian village dogs, with ratios of 1.186–1.217 for HXH and 1.195–1.231 for NGD (Supplementary Table 17).

We then calculated the expectation of this ratio using coalescent theory and iterated over possible $\tau_1$ values until the expectation of the ratio fell into the observed confidence interval. While our estimates of divergence times are in units of expected numbers of mutations, we can use the age of our ancient samples to calibrate the resulting divergence time in years. We used the age of the HXH sample to set an upper bound for the yearly mutation rate $\mu$, as the sample must be younger than the time in years since divergence of HXH and modern European dogs. Given that the sample is ~7,000 years old, we infer that an upper bound for $\mu$ is $5.6 \times 10^{-9}$ per generation (assuming a 3-year generation time, with a 95% CI for the upper bound of $3.7 \times 10^{-9}$ to $7.4 \times 10^{-9}$, Supplementary Fig. 25). This upper bound, which represents the highest mutation rate potentially compatible with the age of our samples, is consistent with the rate of $\mu = 4 \times 10^{-9}$ per generation suggested by both Skoglund et al.[28] and Frantz et al.[12], two rates also calibrated by ancient samples. When we calibrate $\tau_1$ using this mutation rate, we estimate a value of ~6,500–12,900 years for HXH and ~6,400–12,600 years for NGD.

From the G-PhoCS analysis, we further estimated that modern European and Indian village dogs diverged ~13,700–17,900 years ago, both of which diverged from Southeast Asian dogs ~17,500–23,900 years ago as a basal dog divergence event. Finally, we estimated the dog–wolf divergence time to be 36,900–41,500 years ago (Fig. 5b). We note that, though in line with previous studies[7,26], our estimates of east–west dog divergence are much older than those reported in Frantz et al.[12] (6,000–14,000 years ago). While we use a Bayesian approach with G-PhoCS to infer divergence times, Frantz et al.[12] rely on the multiple sequentially Markovian coalescent (MSMC) approach, the performance of which is strongly dependent on the accuracy of genomic phasing[29].

**Functional variants associated with domestication**. As a result of the domestication process, specific portions of dog genomes have significantly differentiated from wolves[30]. To determine the domestication status of the three Neolithic dogs, we assessed haplotype diversity at candidate domestication loci. Using only breed dogs and wolves, a previous study identified 36 candidate domestication loci[30] (Supplementary Table 18). However, our analysis of a more diverse sample set that includes village dogs confirms only 18 of these loci as putative domestication targets, the remainder are likely associated with breed formation (Supplementary Table 19 and Supplementary Note 14). HXH appeared homozygous for the dog-like haplotype at all but one of these 18 loci, and thus was often indistinguishable from most modern dogs. The younger NGD appeared dog-like at all but two

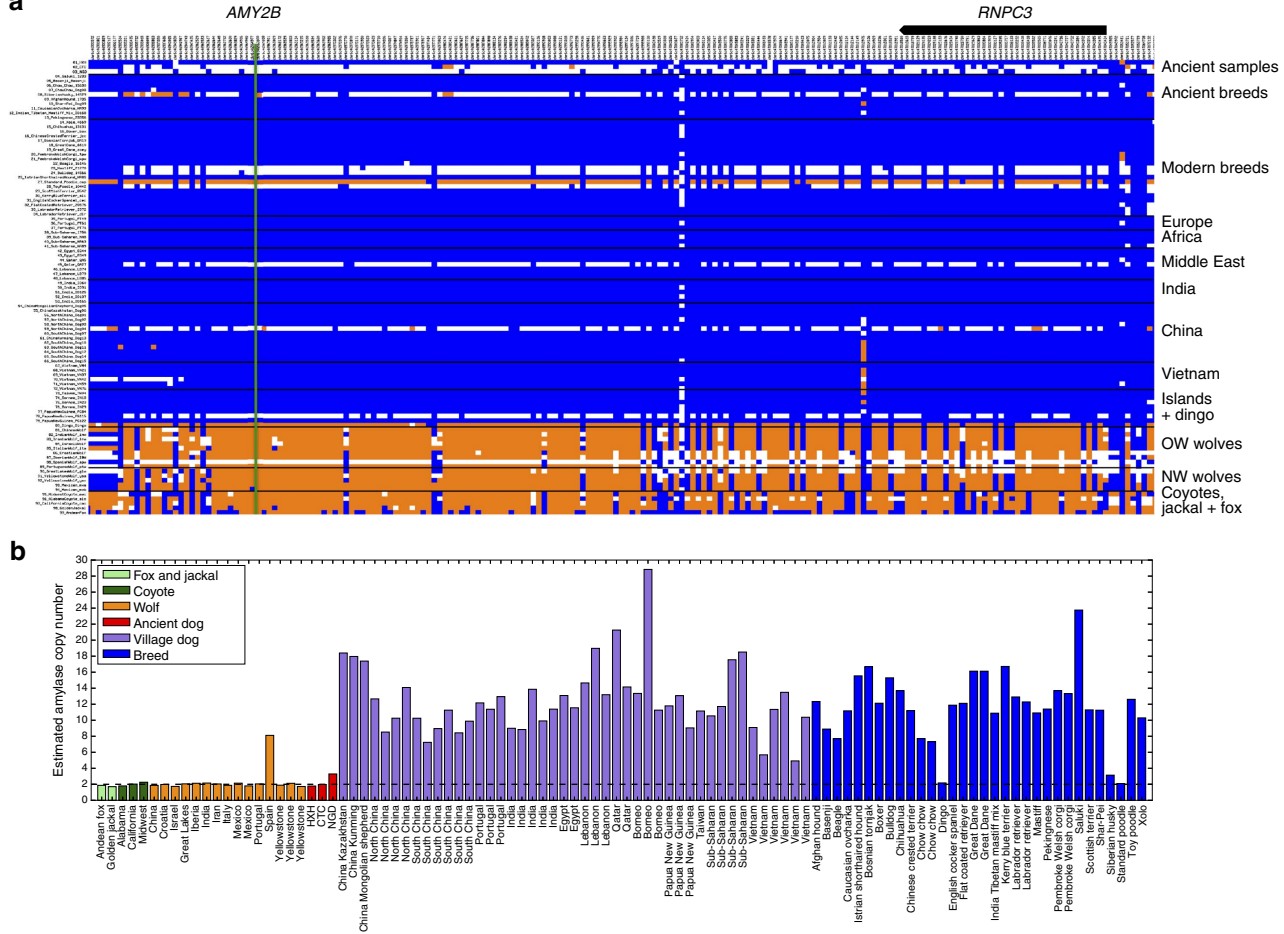

**Figure 6 | Haplotype and copy number variation at the amylase 2B (AMY2B) locus. (a)** Genotype matrix of selected sites within $F_{ST}$-derived domestication locus 12 (chr6: 46854109-47454177)[30]. SNP genotypes are represented as either homozygous for the reference allele (0/0; blue), heterozygous (0/1; white) or homozygous (1/1; orange) for the alternate allele. The positions of *AMY2B* (green line) and *RNPC3* (model above) are indicated. **(b)** Read-depth-based estimation of *AMY2B* copy number for the Andean fox (light green), golden jackal (light green), coyotes (dark green), wolves (orange), ancient samples (red), village dogs (purple) and breed dogs (blue). Dashed line indicates diploid copy number of two.

loci. CTC, however, was heterozygous for the wolf-like haplotype at six loci, compatible with its increased wolf ancestry described above.

The Neolithic saw drastic changes in human culture and behaviour, including the advent of agriculture, resulting in a shift towards more starch-rich diets. Elevated *AMY2B* copy number, which is associated with increased efficiency of starch metabolism, has often been suggested to be a strong candidate feature of domestication, even though *AMY2B* copy number is known to vary widely in diverse collections of modern wolves and breed dogs[26,31,32]. Although the dog haplotype is present in all three Neolithic samples at this locus (Fig. 6a), none showed evidence for the extreme copy number expansion of *AMY2B* (Fig. 6b). On the basis of read depth, we estimate that CTC and HXH carried two copies of the *AMY2B* gene while NGD carried three copies, not two as previously reported[12] (Supplementary Note 14). Analysis of the full sample set of canines shows a bimodal distribution of copy number, with most modern dogs having >6 *AMY2B* copies, while few carry 2 or 3 copies[32]. This dynamic and extreme copy number increase is presumed to be the result of a tandem expansion of the *AMY2B* gene[30]. Further analysis of NGD read-depth profiles has revealed the presence of a larger, ~2 megabase segmental duplication encompassing the *AMY2B* gene locus on chromosome 6 and extending proximally towards

the centromere (Supplementary Fig. 26). This duplication is present in 11 of the analysed modern dog samples and appears to be independent of the extreme copy number expansion of the *AMY2B* gene itself (Supplementary Note 14).

## Discussion

This paleogenomic study provides several new insights into the history of dog domestication in Europe. We find strong evidence for genetic continuity from the Palaeolithic into the Neolithic, and, to some extent, the present. In addition, we do not find any evidence of a now extinct European Palaeolithic dog population contributing to a genetically distinct dog population from either the Early or End Neolithic, and therefore our results do not support the hypothesis of a large population replacement from East Asia during this era. Instead, we find that NGD is genetically very similar to HXH, with both possessing ~70–80% modern European-like ancestry. In addition, CTC most likely directly descended from a population represented by HXH, pointing to some genetic continuity throughout the Neolithic (over 2,000 years) in Central Europe.

However, the admixture events observed in European Neolithic dogs but not in most modern dogs (and even then to a lesser extent) from the same region suggest some degree of population

structure on the continent during that period. This is further reflected by HXH and NGD carrying both Southeast Asian ancestry but lacking the ancestry shared between CTC and modern Middle Eastern, Central and South Asian village dogs, even though NGD and CTC are contemporaneous (4,800 and 4,700 years old, respectively). It is likely that under this scenario of population structure, a subpopulation distinct from that of HXH, CTC and NGD eventually became dominant in modern European dogs, which may explain the observed mtDNA turnover from haplogroup C to A, especially if this subpopulation also passed through a strong bottleneck. Additional support for population structure comes from the clustering of all the ancient samples within C1 into a sub-haplogroup distinct from that of modern dogs, while it is also noteworthy that non-C haplogroups, including A, are more apparent in Southeast Europe in the archaeological record[12].

CTC shows similar admixture patterns to Central Asian and Middle Eastern modern dog populations. Considering that the age of the samples provides a time frame (between 7,000 and 5,000 years ago) for CTC to obtain its unique ancestry component, and that the cranium was found next to two individuals associated with the Neolithic Corded Ware culture, we speculate that this component was derived from incoming populations of dogs that accompanied steppe people migrating from the East[13].

Analyses incorporating admixture in their model show a significant proportion of modern Indian-like ancestry in CTC. However, in addition, there is a potential wolf-like component observed from our NGSadmix and Spacemix analyses, as well as a Southeast Asian component that appears in all three Neolithic dogs. Given such a complex picture of admixture, with four potential sources that must be inferred from a single genome, it is perhaps unsurprising that different methods demonstrate variability in their inferred Indian-like admixture proportions (from 25% in NGSadmix up to 69% in ADMIXTUREGRAPH). We hope that more genomes from Central Europe from this era will help clarify this complicated picture of admixture in the future.

Our older estimate of the 'east–west' divergence time of ∼17,500–23,900 years ago negates the need to invoke a hypothesis of dual dog origins suggested by Frantz et al.[12]. The genomic continuity we see between our 7,000-year old HXH, the ∼5,000-year old NGD and modern European samples implies that if there was any kind of population replacement, it must have occurred before the Neolithic (and perhaps much earlier given the matrilineal continuity between HXH, CTC and Bonn–Oberkassel). However, unlike the propositions of Frantz et al.[12], such a replacement would necessarily be independent of the observed mtDNA turnover of C–A lineages, as this appears to have occurred at least 2,000 years after the end of the Neolithic (that is, separated by at least 4,000 years).

We also estimated the dog–wolf divergence time to be 36,900–41,500 years (Fig. 5b), which is consistent with predictions from the ancient Taimyr wolf genome[28]. As domestication must have occurred subsequent to the dog–wolf divergence and before Southeast Asian dog divergence (∼17,500–23,900 years ago; Fig. 5b) our results provide an upper and lower bound for the onset of dog domestication, between ∼20,000 and 40,000 years ago. To date, Southeast Asia, Europe, the Middle East and Central Asia have all been proposed as potential locations for the origin of dog domestication based on modern genomic data, archaeological evidence and ancient mitochondrial lineages[5,7,9,33]. While our analyses of three Neolithic genomes from Europe have helped narrow the timing of domestication, they are neither old enough nor do they have the broad geographic distribution necessary to resolve this debate. Nonetheless, our work does make clear that population structure and admixture have been a

prominent feature of dog evolution for a substantial period of time. Population genetic analyses based only or primarily on modern data are unlikely to account for such complexity when modelling dog demographic history and therefore paleogenomic data from Upper Palaeolithic remains throughout Eurasia will be crucial to ultimately resolve the location(s) of dog domestication.

Enhanced starch digestion through extreme AMY2B copy number expansion has been postulated to be an adaptation to the shift from the carnivorous diet of wolves to the starch-rich diet of domesticated dogs[30]. Although none of the German Neolithic samples carries the copy number expansion of the AMY2B gene associated with starch digestion, we find that this gene is present in three copies in NGD, though this is due to a large segmental duplication that is shared with multiple modern dogs, an event separate from the tandem AMY2B duplications. This suggests that the initial selection at this locus may have been independently driven by some factor other than AMY2B copy number. The absence of the extreme AMY2B copy number increase in these ancient samples indicates that the selective sweep associated with AMY2B expansion must have occurred well after the advent of agriculture and the Neolithic in Europe. This is consistent with recent findings that AMY2B copy number is highest in modern dog populations originating from geographic regions with prehistoric agrarian societies, and lowest from regions where humans did not rely on agriculture for subsistence[34] and supports the claim that the expansion occurred after initial domestication (possibly after the migration of dingoes to Australia 3,500–5,000 years ago)[34]. A similar pattern has been observed in humans, where alleles associated with lactase persistence in Europe did not rise to significant frequencies until at least the Bronze Age, that is, 3,000 years after the introduction of pastoral livestock[35].

Overall, our findings reveal a history of domestic dogs as intricate as that of the people they lived alongside. The inference of complex patterns of gene flow is challenging, or even impossible, when only modern samples are studied. Therefore, the acquisition of a broader set of ancient samples, including ancient representatives from Central and Southeast Asia, and the Middle East will be crucial to further clarify the details of dog domestication and evolution.

## Methods

**Archaeological background.** For the HXH sample, a single petrous bone was identified in the internal ditch structure of Herxheim, an Early Neolithic site in Germany discovered in 1996, which contained archaeological material from the Linearbandkeramik culture. Herxheim contains a significant amount of faunal remains, including >250 remains from dogs that constitute the largest bone series of Early Neolithic dogs in Western Europe. A $^{14}$C dating of 5,223–5,040 cal. BCE (95.4%) was estimated for the bone (Mams-25941: 6186 ± 30, calibrated with OxCal 4.2 (ref. 36) using the IntCal13 calibration curve[37]).

For the CTC sample, the entire cranium of a dog was found in the Kirschbaumhöhle (Cherry Tree Cave) in the Franconian Alb, Germany[14] (Supplementary Fig. 27). The cave was discovered in 2010 and contains human and animal remains from at least six prehistoric periods. CTC was an adult dog demonstrating morphological similarity to the so-called Torfhund (Canis familiaris palustris), and was found close to two human skulls dated to the early End Neolithic (2,800–2,600 cal. BCE ). A $^{14}$C dating of 2,900-2,632 cal. BCE (95.4%) was estimated for the cranium (Erl-18378: 4194 ± 45, calibrated with OxCal 4.2 using the IntCal13 calibration curve). See Supplementary Note 1 for more details.

**DNA isolation and screening.** For the HXH sample, the petrous part of the temporal bone of sample HXH was prepared in clean-room facilities dedicated to ancient DNA in Trinity College Dublin (Ireland). DNA extraction was performed using a Silica column method as described in MacHugh et al.[38]. Two genomic libraries were prepared as described in Gamba et al.[39]. Screening of one library via an Illumina MiSeq run and mapping against various reference genomes demonstrated that reads for this sample mapped almost exclusively to the CanFam3 genome, revealing that it was a canid. Blank controls were utilized throughout. See Supplementary Note 2, Supplementary Figs 28 and 29 and Supplementary Tables 20 and 21 for more details.

For CTC, sample preparation was conducted in dedicated ancient DNA facilities of the Palaeogenetics Group at Johannes Gutenberg-University Mainz under strict rules for contamination prevention as described in Bramanti et al.[40]. DNA was extracted independently twice from the petrous bone using a phenol–chloroform protocol[41]. A total of four double-indexed genomic libraries were prepared as described in Hofmanová et al.[17]. One library was screened for endogenous DNA content via Illumina MiSeq sequencing, with 61.5% of reads mapping to CanFam3. Blank controls were utilized throughout. See Supplementary Note 3, Supplementary Fig. 30 and Supplementary Table 22 for more details.

**Genome sequencing and bioinformatic processing.** Combinations of various genomic libraries from each ancient sample (CTC and HXH) were sequenced on two lanes of an Illumina HiSeq 2500 1TB at the New York Genome Center (NYGC) using the high output run mode to produce $2 \times 125$ bp paired-end reads. Reads were trimmed, merged and filtered using a modified version of the ancient DNA protocol described by Kircher[42]. Merged reads were then mapped using BWA aln[43] to a modified version of the CanFam3.1 reference genome containing a Y chromosome. Duplicate reads were identified and marked using PICARD MarkDuplicates, resulting in a mean coverage for both samples of $2 \times 125$ bp. In addition, the mean coverage for the X and Y chromosomes was $\sim 5 \times$ for both samples, indicating they were males. Mean fragment length for both samples ranged from 60 to 70 bp. Postmortem degradation effects were assessed using MapDamage_v1.0 (ref. 15), revealing extensive 5′ C>T and 3′ G>A damage. Single-ended reads for NGD extracted from a BAM file containing all mapped reads were processed using the same pipeline. See Supplementary Note 4 for more details.

Genotype likelihood estimation and genotype calling for all three ancient samples were performed using a custom caller that takes into account postmortem damage patterns identified by MapDamage based on the model described in Hofmanová et al.[17]. Briefly, damage patterns with respect to read position are fit with a Weibull distribution of the form $a \times \exp(-(x^c) \times b)$, where $x$ is the proportion of damaged C>T or G>A bases at a particular position along the read (unlike Hofmanová et al.[17], we find a slightly better fit with a Weibull than when assuming exponential decay) (Supplementary Fig. 31). Any site with $<7\times$ coverage was reported as missing. In addition, any position where the highest likelihood is a heterozygote must have a minimum Phred-scaled genotype quality of 30 or the next highest homozygote likelihood genotype was chosen instead. The code is available at https://github.com/kveeramah/aDNA_GenoCaller. This protocol substantially decreased the overrepresentation of C>T and G>A sites identified by GATK UnifiedGenotyper[44], which does not account for postmortem damage. In addition, base calls with a quality score $<15$ and reads with a mapping quality $<15$ were not included during genotype calling. Base calls with a quality score $>40$ (which can occur during paired-end read merging) were adjusted to 40. See Supplementary Note 5 and Supplementary Figs 32 and 33 for more details.

**Reference data set.** To construct a genome sequence data set, in addition to the three ancient samples, we examined whole-genome sequence data from 96 modern canids. Additional genomes were generated using Illumina sequencing for a Great Dane and Iberian wolf (SRP073312). We also posted sequencing reads to the SRA for a Portuguese village dog, Chinese Mongolian Shepherd village dog and a Sub-Saharan African village dog (SRP034749). All remaining genome data were acquired from previously published data sets deposited on SRA. As above, reads for all modern canids were aligned to CanFam3.1 using BWA, followed by GATK quality score recalibration, and genotype calling using HaplotypeCaller[44]. These data were supplemented with genotype data for six canids from Freedman et al.[26] (basenji, dingo, golden jackal, Croatian wolf, Israeli wolf and Chinese wolf). We generated three different call sets with different ascertainment schemes. Call set 1 includes all variants from both ancient and contemporary genomes, representing the most comprehensive set of variants, but may show biases due to differences in coverage among sample sets. Call set 2 only includes variants discovered in the three ancient genomes. Call set 3 only includes sites discovered as variable in New World wolves, and is the primary call set utilized for most analyses. See Supplementary Note 6 for more details.

To construct a SNP array data set, canine SNP array data sets were obtained from Shannon et al.[9] and Pilot et al.[18]. Genotypes were also supplemented by data from the six canids reported in Freedman et al.[26].

**Statistical analysis.** The average sequencing depth for mtDNA was $179 \times$, $208 \times$ and $170 \times$ in the CTC, HXH and NGD samples, respectively. Ancient sample mtDNA consensus sequences were aligned to the canid alignment from Thalmann et al.[5], which contain whole mtDNA genomes for both modern and ancient canids. A NJ tree was built with a TN93 substitution model (500 bootstraps) using MEGA 6.06 (ref. 45). A further NJ tree was built with additional C1 and C2 samples from Duleba et al.[19]. See Supplementary Note 7 for more details.

NJ trees were constructed for the whole-genome SNP set using the ape R package[46] using distance matrices based on the metric of sequence divergence from Gronau et al.[25]. One hundred bootstrap replicates were generated by dividing the genome into 5 cM windows and sampling with replacement to determine node support. See Supplementary Note 8 for more details.

PCA was performed on both the SNP array data set and genome SNP call set 3 using smartpca, part of the EIGENSOFT package version 3.0 (ref. 47). Both diploid and pseudo-haploid genotype calls with and without C<>T and G<>A SNPs (the most likely sites to undergo postmortem damage) were used to construct the PCA, but little difference was found among these analyses. SpaceMix[24] was used to create a geogenetic map and infer potential long-distance admixture events across this map using the SNP array data, allowing only SNPs separated by at least 100 kb and no more than five individuals per population. Multiple runs were performed with 10 initial burn-ins of 100,000 generations and a final long run of 10,000,000 generations. ADMIXTURE (v. 1.22)[48] was used to perform an unsupervised clustering analysis on the SNP array data for the ancient dogs and a subset of 105 modern dogs that provided a global representation of dog structure, while NGSadmix[49] was used to perform a similar analysis for the genome SNP data while taking into account genotype uncertainty by examining genotype likelihoods. Cross validation was performed for the ADMIXTURE analysis to identify the most appropriate number of clusters, K. See Supplementary Note 9 and Supplementary Figs 34 and 45 for more details.

Outgroup-f3 statistics were used to assess relative genetic drift between ancient and modern dogs. This method has been used previously in ancient DNA studies to investigate how modern populations are genetically related to an ancient sample[17,21,35,50]. Assuming a simple three population model with no post-divergence gene flow, where population C is an outgroup to A and B, the value of this statistic will reflect the amount of shared drift between A and B relative to C. If one population (for example, B) is kept constant, in this case an ancient dog, then introducing different populations to represent A will provide relative estimates of genetic similarity with B (note this makes no assumptions with regard to the complexity of the demographic history that connects populations A and B). Outgroup-f3 statistic maps were created using the R packages ggplot2 and maps using the public domain Natural Earth data set (http://www.naturalearthdata.com). D-statistics were used to identify potential ancient dog–wolf admixture and f4-ratio tests to estimate dog–dog and dog–wolf admixture proportions were calculated using Admixtools[22]. See Supplementary Note 10 and Supplementary Fig. 46 for more details. Both MixMapper[23] and ADMIXTUREGRAPH[22] were used to perform model-based inference of specific admixture events involving the three ancient dogs. MixMapper was performed on both the SNP array and whole-genome SNP data sets, whereas ADMIXTUREGRAPH was performed on the whole-genome data set only. Significance was assessed using a weighted block jackknife procedure for all five analysis types. Genetic map positions for each SNP used in these analyses were inferred from Auton et al.[51]. See Supplementary Note 11 and Supplementary Figs 47 and 48 for more details.

G-PhoCS[25] was used to estimate divergence times, effective population sizes and migration rates for various modern dog and wolf combinations using sequence alignments from 16,434 'neutral' loci previously identified in Freedman et al.[26] after LiftOver from CanFam3 to CanFam3.1. NJ trees were constructed to inform the topology of population divergence. A total of 500,000 Markov chain Monte Carlo (MCMC) iterations were found to be sufficient for convergence for our data, with the last 200,000 used to estimate posterior distributions. We then developed a numerical approach based on coalescent theory to predict the ratio of shared derived sites between HXH/NGD and European village dogs versus Indian and European village dogs given a particular divergence time of HXH/NGD in units of expected numbers of mutations ($P_1$). Our expectation was conditioned on the following parameter estimates from G-PhoCS: $N_e$ for European/Boxer ancestral population ($\theta_1$), $N_e$ for European/Indian ancestral population ($\theta_2$), time of divergence for Europe and Boxer ($P_0$), time of divergence for Europe/India ($P_2$) and time of divergence for Europe-India/Asia ($P_3$) as well as the percentage of HXH that is made up of Asian admixture ($\alpha$) from the f4-ratio analysis (Supplementary Table 23). Confidence intervals were estimated by resampling G-PhoCS parameters from their posterior distributions and finding predicted-derived allele sharing ratios that were within a range determined for the observed data by a weighted jackknife resampling approach. See Supplementary Notes 12 and 13, Supplementary Figs 49–55 and Supplementary Table 23 for more details.

Coordinates of 30 putative 'domestication loci' were obtained from Axelsson et al.[30] and lifted over from CanFam2.0 to CanFam3.1 coordinates. Call set 1 SNPs within each window were extracted from the ancient samples and our genome sequence data set. Eigenstrat genotype file formats were generated per window using convertf from the EIGENSOFT package[52] and custom scripts were used to convert the genotype files into matrix formats for visualization using matrix2png[53] using a filtered subset of SNPs (minor allele frequencies between 0.05 and 0.49) for easing visualization of the matrices. NJ trees were estimated for each window with the full SNP set using the same methods as the whole-genome tree estimation (see above). Altogether, the haplotypes of the three ancient samples were classified as either dog or wolf-like for 18 matrices that showed clear distinction between dog and wild canid haplotypes based on average reference allele counts calculated per window. See Supplementary Note 14 and Supplementary Figs 56–50 for more details.

Genomic copy number at the amylase 2B locus was estimated from read depth as previously described[54,55]. Specifically, reads were split into non-overlapping 36 bp fragments and mapped to a repeat-masked version of the CanFam3.1 reference using mrsFAST[56], returning all read placements with two or fewer substitutions. Raw read depths were tabulated at each position and a loess

correction for local GC content was calculated utilizing control regions not previously identified as copy number variable. The mean depth in 3 kb windows was then calculated and converted to estimated copy number based on the depth in the autosomal control regions. See Supplementary Note 14 for more details.

PCA, divergence time between eastern and western dogs and demographic models tested with ADMIXTUREGRAPH are compared with Frantz et al.[12] and discussed in Supplementary Note 15.

**Data availability.** Sequencing data are available from the NCBI sequence read archive (SRA) database under accession numbers SRS1407451 (CTC) and SRS1407453 (HXH), and mitochondrial genomes are available in GenBank under accessions KX379528 and KX379529, respectively. Code generated to call variants in the ancient samples is available at: https://github.com/kveeramah/aDNA_GenoCaller.

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

## Acknowledgements

We thank Dan Bradley for his help obtaining the HXH specimen. We thank Walter Eanes and Douglas Futuyma for their comments on the manuscript, Dorina Twigg for the processing of canine copy number variation data, Nick Patterson for providing advanced access to the latest version of Admixtools, Vida for his thoughts, the NYGC for their assistance in the sequencing, Valeria Mattiangeli for performing initial Miseq sequencing on HXH, Christian Sell for his assistance with the raw data analysis pipeline used for the CTC Miseq data, and the Musée zoologique de la Ville de Strasbourg for hosting the team of archaeozoology. The Kidd Lab is supported by NIH Grant R01GM103961 and A.L.P. is supported by T32HG00040. A.S. and K.D. are supported by

the EU: CodeX Project No: 295729. Laura Botigué is supported by the Beatriu de Pinós Fellowship, from Generalitat de Catalunya. S.G. was supported by a Boehringer Ingelheim Fonds Travel award.

## Author contributions

T.S., A.Z. and R.A. provided the archaeological material. A.S., S.G., K.D. and M.U. performed the ancient DNA lab work and screening. L.R.B., S.S., A.L.P., M.O., A.M.T., D.B., J.M.K. and K.R.V. performed the downstream bioinformatics and population genetic analysis. L.R.B., J.B. and K.R.V. conceived the study. L.R.B. and K.R.V. wrote the paper.

## Additional information

**Competing interests:** The authors declare no competing financial interests.

