## [Peer Review File · Nature Communications]

Reviewer #1 (Remarks to the Author):

This paper presents the sequence analysis of nuclear (and mitochondrial) genomes from three Neolithic European dogs, adding two central European (one early and one late Neolithic) dogs to the already sequenced Neolithic Irish dog from Frantz et al. (2016).

This is an important study because it tests the recent suggestion (by Frantz et al.) of a replacement of European dogs in the late Neolithic, indicating that domestic dogs had a dual origin from wolf, one in Eastern Asia and one in Europe. This assertion was based on quite shaky data (for example presuming very exact dating), and goes against most previous datasets, which have suggested that dogs had a single geographical origin. It is therefore of importance that the genetic history of ancient European dogs is studied in more exact detail.

Contrary to the previous claims by Frantz et al. this paper shows quite convincingly that there is genetic continuity from the earliest Neolithic all the way to modern European dogs, strongly contradicting the theory that late Neolithic European dogs were replaced.

Given that a controversial scenario can be convincingly rejected, and that it shows a convincing picture of the early history of domestic dog in Europe this paper will have a great impact within the field, and will also clearly be of great general interest.

The design and analyses are quite solid. The two samples are strategic, in the beginning and end of Neolithic, and in central Europe (the previous NGD being from a the peripheral island of Ireland), thus knitting together the whole Neolithic period with modern dogs, as well as covering large parts of Europe. Therefore, this is the first really comprehensive study of genome diversity in Neolithic European dogs.

The authors have performed an impressive amount of diverse analyses. An important aspect is the parallel analysis of genome sequence and SNP-data (larger but affected by ascertainment bias, specifically expected to be biased in inflating European diversity), giving largely congruent pictures. This serves as a powerful test of the analyses (as well as those in Frantz et al.). I do not find any clear problems with the analyses (while importantly pointing out several mistakes in Frantz et al.)

In addition to the results concerning the genetic ancestry relations between Neolithic and modern European dogs, the study also gives a rich material concerning timing of different steps in dog history, e.g. dog-wolf split and East-West split, as well as indications of admixtures and migrations.

Finally, the results concerning the 18 "domestic dog specific" genes and the non-expansion of *AMY2B* also adds important knowledge about the earliest evolution of the domestic dog.

Thus, this study gives a large amount of new data concerning the earliest evolution of the domestic dog, which will be of great importance for the field.

I have one major comment which I think must be addressed:

I appreciate very much the authors' precise and cautious interpretation of the data. Unlike many previous studies about dog origins, the authors refrain from making too farfetched conclusions.

However, this is a study about the origins of dogs, arguing strongly against a dual origin of dogs, instead claiming a single origin. But, the authors do not discuss where this single origin may have occurred, and this is therefore a question hanging in the air.

The authors must address the question where dogs actually originated. Given the data, is there any strong (or weak) indications for a specific geographical region? Or can any regions be excluded? In the case the authors don't think the data gives any clear indications, they should say so, and indicate why, and which further analyses will be needed to come any nearer the answer to this question.

In conclusion, I believe this study will serve as one important pillar in the final elucidation of the origins of the domestic dog. I am therefore obviously positive to this study, I think it is important for the field, and that it is clearly of sufficient importance and interest for the field as well as general community to be accepted for publication in this journal.

Reviewer #2 (Remarks to the Author):

Botigue et al presented two ancient European dog genomes from early and late Neolithic periods. Population genetic analyses revealed genomic continuity rather than population replacement found earlier in the Frantz et al 2016 study. The genetic pattern didn't support the dual origin of dog domestication proposed by Frantz et al. These two ancient genomes share partial list of selective sweeps with modern breeds, stratifying the selective process of dog domestication. The amount of data and analyses presented are quite extensive, and many conclusions are quite interesting. This paper could be a very nice contribution for Nature Communications. However, the paper is not very well organized and many pieces are not coherent with each other. The authors seem to be quite new to the field and very few connections are made to the known discoveries from the field. I have several concerns:

a) An earlier study of Wang et al 2016 (Cell Research) found that dogs from northern China have extensive European/Asian admixture. From the admixture analysis presented in this paper, this is also true in both HXH and NGD. It is curious whether these admixture events are related or not. In other words, are they descending from the same population?

Following up this point, the ancient dogs (HXH/NGD/CTC) are sister clades to Modern European dogs and are not direct ancestors of European dogs (all have Asian admixture). Then, where did the pure European component come from and how did they become so dominant in Europe?

The authors may perform f_4 -ratio test by using the accordant phylogenetic topology based on whole genome analysis, when they detected the Southeast Asian-like gene flow into HXH and NGD. They used East Asia village dogs as A and B, and used Portugal village dogs as C (Table S10.4.1), which is a sister clade of the common ancestor of A and B. However, in the Neighbor-joining tree based on

pairwise sequence divergence from whole genome data, the East Asian village dogs are basal to all other dogs.

In the results of PCA (fig 1A), CTC is clustered with India dogs and far away from Europe group (include NGD and HXH). Moreover, the best model fit of ADMIXTUREGRAPH (fig1a) show there is 69% component from India dogs in CTC. However, in the Neighbor-joining tree and NGSadmix (K=4), CTC is together with Europe group (include NGD and HXH). The authors have not discussed the inconsistency.

Authors may add South China dogs or Southeast Asia dogs in ADMIXTUREGRAPH analysis, since there is 8-15% migration from South China dogs to the common ancestor of CTC, HXH and NGD, and South China dog is an ancient clade in the NJ-tree.

b) In the “modern canid reference datasets”, it is unknown why the authors curated the SNP lists using the ascertainment scheme implemented in the paper. Even though the authors stated the observation that the primary conclusions are robust to this type of ascertainment, the scheme is not very natural. The authors should be more explicit about the reasons for this implementation. It is also a bit unclear what analyses are using this SNP set (e.g. also used for Mixmapper and ADMIXTUREGRAPH?).

c) The materials surrounding the genetic distances of these two individuals to other canids are slightly disorganized (started with Figure 2A/2B, then Figure 3 and subsequently figure 2C). It might be good to put Figure 2A/2B together with Figure 3. Subsequently, move Figure 2c to Figure 4.

d) Figure 2 is poorly drawn (e.g. Figure 2A and 2B are hardly readable. The colors are very overlapping). This also applies to Figure 4A. The symbols are very poorly explained.

e) The “demographic model and divergence time”,

i) Is 0.5247 the expected number of substitutions per 1kb bases?

ii) The mutation rate is a quite difficult metric now. There have been many mutations rate used (e.g. 4.0×10^{-9} (estimated from Skoglund et al), 6.6×10^{-9} (from Wang et al 2016/2013) and 1×10^{-8} (Freedman et al and the dog reference genome paper). The authors reached a conclusion similar to Wang et al estimate (both in terms of divergence time/0.2786 and mutation rate 6.0×10^{-9} per site per generation). But the authors lean towards 4.0×10^{-9} and ended up reaching a very ancient divergence time for dog domestication. I am not sure this is well supported by the genetic and archaeological data. The authors should be aware of these uncertainties in light of the previous findings from the field.

Reviewer #1:

1) The authors do not discuss where this single origin may have occurred, and this is therefore a question hanging in the air. The authors must address the question where dogs actually originated. Given the data, is there any strong (or weak) indications for a specific geographical region? Or can any regions be excluded? In the case the authors don't think the data gives any clear indications, they should say so, and indicate why, and which further analyses will be needed to come any nearer the answer to this question.

We thank the reviewer for this thoughtful summary of our study. In answer to their main critique, we do not feel that three Neolithic samples from Europe represent the breadth of sampling required, both with regard to age or sampling location, to make definitive statements regarding the geographic origins of dogs. We predict that paleogenomic data from the Paleolithic era, particularly from both Europe and Asia, will be crucial for this in the future. However, we agree that inclusion of additional statements regarding the types of data required to reconcile competing theories about the geographic origins of dogs would be an important addition to the manuscript. We have therefore address this matter as a new paragraph in the Discussion section (pg 18, para 3) that addresses these points. Please find the paragraph below for your convenience:

“To date Southeast Asia, Europe, the Middle East and Central Asia have all been proposed as potential locations for the origin of dog domestication based on modern genomic data, archaeological evidence and ancient mitochondrial lineages^{8,9,21,35}. While our analyses of three Neolithic genomes from Europe have helped narrow the timing of domestication, they are neither old enough nor do they have the broad geographic distribution necessary to resolve this debate. Nonetheless, our work does make clear that population structure and admixture have been a prominent feature of dog evolution for a substantial period of time. Population genetic analyses based only or primarily on modern data are unlikely to account for such complexity when modeling dog demographic history and, therefore, paleogenomic data from Upper Paleolithic remains throughout Eurasia will be crucial to ultimately resolve the location(s) of dog domestication.”

Reviewer #2:

1) An earlier study of Wang et al 2016 (Cell Research) found that dogs from northern China have extensive European/Asian admixture. Following up this point, the ancient dogs (HXH/NGD/CTC) are sister clades to Modern European dogs and are not direct ancestors of European dogs (all have Asian admixture). Then, where did the pure European component come from and how did they become so dominant in Europe?

We note that nowhere in our manuscript do we describe a “pure European component”. Instead we emphasize in the original text that descriptions of dog populations as clean bifurcating clades will not always be completely realistic, especially when describing ancient and modern European dogs as sister clades (pg 9, paragraph 2: “We note that this is a simplistic bifurcating model of what may have been more complex European geographic structuring and long term Eurasian dog gene flow.”). Instead, this acts as a useful model for making certain inferences about general patterns, like substantial genetic substructuring (and thus implied gene flow amongst subpopulations) within the European continent until recently.

To better clarify our model to the reviewer, we are proposing based on our various analyses that the modern European component arose when the ancestor of this population diverged from Asian dogs ~20kya, and Indian dogs ~15ya (see Fig 5b), though where geographically these splits occurred cannot be determined based on our data alone. All three Neolithic dogs are genetically very close to this modern European lineage (particularly HXH and NGD), and thus the general “pure” European component must have been in Europe by at least 7kya. But there would have been subsequent admixture from dogs that closely resemble modern Southeast Asian dogs into Europe at some point by the end of the Neolithic. This admixture may have had differential effects on an already structured European population, as observed by varying amount of such Southeast Asian components in HXH, NGD and even some modern European dogs (see figure S8.3.2). We note that we included a statement in the original Results section (pg 6, paragraph 3) saying “*all three ancient genomes possess a significant ancestry component that is present in modern Southeast Asian dogs. This component appears only in a minority of modern European village dogs at very low levels.*” to draw attention to the fact that even modern dogs contain some Southeast Asian ancestry. In response to the reviewers concerns, we have also now slightly modified the section of the discussion where we discuss European population structure (pg 13, paragraph 2) to re-emphasize this point to the reader.

“However, the admixture events observed in European Neolithic dogs but not in most modern dogs (and even then to a lesser extent) from the same region suggest some degree of population structure on the continent during that period. This is further supported by HXH and NGD carrying both Southeast Asian ancestry but lacking the ancestry shared between CTC and modern Middle Eastern, Central and South Asian village dogs, even though NGD and CTC are contemporaneous (4,800 and 4,700 years old, respectively). ***It is likely that under this scenario of population structure, a different subpopulation eventually became dominant in modern European dogs, which may explain the observed mtDNA turnover from haplogroup C to A, especially if this subpopulation also passed through a strong bottleneck.*** Additional support for population structure is the differential clustering of the ancient samples within C1 into a sub-haplogroup distinct from that of modern dogs. Though sample sizes are low, it is noteworthy that non-C haplogroups, including A, are more apparent in Southeast Europe in the archaeological record.”

Finally, we would like to note as well that we also observed European-like ancestry in northern (and even some southern) Chinese dogs (see NGSadmixture and ADMIXTURE plots), but do not comment on this in the main text as we believe it had already been well established by Wang et al. and added little to our analysis of the Neolithic dogs.

2) The authors may perform f₄-ratio test by using the accordant phylogenetic topology based on whole genome analysis, when they detected the Southeast Asian-like gene flow into HXH and NGD. They used East Asia village dogs as A and B, and used Portugal village dogs as C (Table S10.4.1), which is a sister clade of the common ancestor of A and B. However, in the Neighbor-joining tree based on pairwise sequence divergence from whole genome data, the East Asian village dogs are basal to all other dogs.

We thank the reviewer for noting this inconsistency in our application of the f₄-ratio test and the known population phylogeny. We have now repeated the f₄-ratio analysis by instead using a South/Southeast Asian dogs as C, Indian dog as A and European dog as B. This leads to a increased estimate of South/Southeast Asian dog ancestry in HXH and NGD (~25%) compared to the previous result and is almost completely concordant with our MIXMAPPER and ADMIXTUREGRAPH analysis (unsurprisingly as these are also based on f-statistics). We also found that the f₄ ratio estimate for Indian dog admixture in CTC was not appropriate due to the complexity of the admixture. We instead use NGSadmix estimate of Indian dog admixture in CTC. We have adjusted our figure (Figure 5b) to reflect this value and changed the relevant values in the main text (pg 7, paragraph 2). In addition, as the result from this f₄ ratio test was used in the mutation rate calibration, we have reperformed this analysis using the new admixture proportion. This results in a slight decrease in the upper bound of the mutation rate from 6.0×10^{-9} (95% CI 4.23×10^{-9} to 7.8×10^{-9}) to 5.6×10^{-9} (95% CI 3.7×10^{-9} to 7.4×10^{-9}). We have adjusted the main text (pg 11, paragraph 1) and supplementary information accordingly.

3) In the results of PCA (Figure 1a), CTC is clustered with Indian dogs and far away from Europe group (include NGD and HXH). Moreover, the best model fit of ADMIXTUREGRAPH (Figure 5a) show there is 69% component from India dogs in CTC. However, in the Neighbor-joining tree and NGSadmix (K=4), CTC is together with Europe group (include NGD and HXH). The authors have not discussed the inconsistency.

A key finding of our analysis is indeed that CTC appears to share some ancestry that is predominantly found in modern Indian dogs (which we refer to as India-like as it is also found in Central Asian and Middle Eastern dogs). All analyses that incorporate some model of admixture show this. We would argue that visually CTC is approximately halfway between European and Indian dogs rather than being “clustered with Indian dogs”, but nonetheless, any Indian-like component inferred from this PCA analysis is clearly significant, as correctly pointed out by the reviewer. CTC also appears to show a clear Indian component in both the NGSadmix and ADMIXTURE clustering analysis, on the order of 25% (so closer to modern Europeans and thus consistent with the Neighbor joining tree analysis). The MixMapper and ADMIXTUREGRAPH analysis also point to other higher values of Indian-like admixture.

What is clear therefore, and what we attempt to emphasize in the paper, is that CTC contains a European component by way of genetic continuity by HXH, as well as an Indian-like component. However, the relative amount of admixture is not clear, and we try not to make strong statements quantifying this. As described in the original supplementary material “*it proved extremely difficult to fit a suitable ADMIXTUREGRAPH for CTC along with other dogs and wolves*” (section 11.2). We believe the case of Indian-like admixture in CTC (especially as pertained to quantifying and fitting the ADMIXTUREGRAPH) is further complicated by an additional wolf component found via the NGSadmix, SpaceMix and domestication analysis.

Overall therefore there is evidence for at least 4 distinct genetic ancestries in CTC (European, Indian, Southeast Asian and wolf), which undoubtedly makes quantifying exact proportions very difficult with just a single genome.

In sum, we agree with the reviewer that there needs to be greater discussion about this issue in the main text. Therefore, we have added a statement noting the challenge in estimating the proportion of Indian-like admixture in the Discussion (pg 13 paragraph 4):

“As well as demonstrating a clear European component and genetic continuity with HXH, all analyses incorporating admixture in their model show a significant proportion of Indian-like ancestry in CTC (we note this component is most predominant in some modern Indian dog populations but is also present in Central Asian and Middle Eastern dogs). However, in addition there is a potential wolf-like component observed from our NGSadmixture and Spacemix analysis, as well as the Southeast Asian component that appears in all three Neolithic dogs. Given such a complex picture of admixture with four potential sources that must be inferred from a single genome, it is perhaps unsurprising that different methods demonstrate somewhat wide variability in their inferred admixture proportions (from 25% in NGSadmixture up to 69% in ADMIXTUREGRAPH). We hope that more genomes from Central Europe from this era will help clarify and better quantify this complicated picture of admixture in the future.”

We note that we have also included the full range of admixture estimates across analyses types in Fig 5, where 5a reflects the ADMIXTUREGRAPH proportion estimate and Figure 5b that of NGSadmixture.

4) Authors may add South China dogs or Southeast Asia dogs in ADMIXTUREGRAPH analysis, since there is 8-15% migration from South China dogs to the common ancestor of CTC, HXH and NGD, and South China dog is an ancient clade in the NJ-tree.

We note that Southeast Asian dogs were included in our original ADMIXTUREGRAPH analysis because, as the reviewer points out, they have made a very significant contribution to the Neolithic dog ancestry. In the main Figure 6a they are represented by Bornean village dogs. We generally avoided using mainland Asian dogs in such analysis as they may contain recent European admixture, but results would likely be very similar.

5) In the “modern canid reference datasets”, it is unknown why the authors curated the SNP lists using the ascertainment scheme implemented in the paper. Even though the authors stated the observation that the primary conclusions are robust to this type of ascertainment, the scheme is not very natural. The authors should be more explicit about the reasons for this implementation. The use of ascertaining SNPs in an outgroup (in this case the North American Wolves) is now a frequently used strategy for population genetic analysis (especially paleogenomics) that rely on only examining changes in allele frequencies between populations (i.e. drift), as one does not need to take into account mutation in any analytical framework and inferences are more robust to variation in read coverage across samples (a typical problem in ancient DNA analysis where endogenous DNA is often rare). Indeed, almost the entire suite of methods developed by Patterson et al. (f3/f4 tests) rely on ascertainment schemes via this strategy. The alternative is to not ascertain SNPs and to incorporate mutation (for example population genetic analysis that use coalescent or diffusion approximations that the reviewer may consider more natural), but when coverage is variable, such an approach can be unreliable for a host of different reasons.

Hence, we use only high coverage genomes with no ascertainment scheme when performing the G-PhoCS analysis.

To better clarify our choice, we have therefore incorporated extra detail to the main manuscript on pg 3, paragraph 4 that better describes the motivation of the ascertainment scheme that we utilized (i.e. ascertainment of variable sites in New world wolves).

“In order to utilize whole genome data with such variable coverage (including our ancient samples), it is important that variable sites are chosen in a manner that will not bias downstream population genetic analysis. One popular approach exemplified by the *f*-statistic analyses of Patterson et al. ¹⁹ is to ascertain variable sites in an outgroup (i.e. such that mutations are known to have occurred in the root of all the populations being analyzed)”

Also more information on the details can be found in Supplementary Material section 6, which describes our motivation and scheme in detail.

6) It is also a bit unclear what analyses are using this SNP set (e.g. also used for MixMapper and ADMIXTUREGRAPH?).

We now specify that ADMIXTUREGRAPH was performed on the whole genome dataset in the main text (pg 7 paragraph 2) “Analysis with ADMIXTUREGRAPH ²⁸ on the whole genome dataset” and methods (pg 20, paragraph 1): “ Both MixMapper ²⁷ and ADMIXTUREGRAPH ²⁸ were used to perform model-based inference of specific admixture events involving the three ancient dogs. MixMapper was performed on both the SNP array and whole genome SNP datasets, whereas ADMIXTUREGRAPH was performed on the whole genome dataset only. ” We also specify it in the figure legend as well. For MixMapper we specify that it was done on both datasets (pg 7 paragraph 2) “We formally modelled these potential admixture events by applying the tree-based framework, MixMapper ²⁷ to both the SNP array and whole genome data”. This information for the other analysis can be found in the main text and the figure captions.

7) The materials surrounding the genetic distances of these two individuals to other canids are slightly disorganized (started with Figure 2A/2B, then Figure 3 and subsequently figure 2C). It might be good to put Figure 2A/2B together with Figure 3. Subsequently, move Figure 2c to Figure 4.

Following the reviewer’s advice, we have split Figure 2 so that the NGSadmixture results are now reported as Figure 4. We agree that this better reflects the order in which analyses are described in the manuscript text and we hope this will ease the reading.

8) Figure 2 is poorly drawn (e.g. Figure 2A and 2B are hardly readable. The colors are very overlapping). This also applies to Figure 4A. The symbols are very poorly explained. We have substantially increased the resolution of the figure and made it larger. The aim of the PCA plot is to capture the general relationships among continents, rather than among specific groups of populations. We also note that this PCA is broadly similar to others previously published, as for instance, Shannon et al. 2015. We have added additional explanations in the legends of Figure 4A, (now figure 5a), to clarify the meaning of the various symbols.

9) The “demographic model and divergence time”,

i) Is 0.5247 the expected number of substitutions per 1kb bases?

We thank the reviewer for catching this typo. This is the expected number of mutations per site (0.5247×10^{-4}). We have clarified the units of this parameter in the main text (pg 10 paragraph 1): “ We found that the (uncalibrated) dog-wolf divergence time in units of expected numbers of mutations per site (0.5247×10^{-4}) was similar to that reported in Freedman³¹; however, our dog divergence time (0.2786×10^{-4}) was younger than the Freedman et al.³¹”.

ii) The mutation rate is a quite difficult metric now. There have been many mutations rate used (e.g. 4.0×10^{-9} (estimated from Skoglund et al), 6.6×10^{-9} (from Wang et al 2016/2013) and 1×10^{-8} (Freedman et al and the dog reference genome paper). The authors reached a conclusion similar to Wang et al estimate (both in terms of divergence time/0.2786 and mutation rate 6.0×10^{-9} per site per generation). But the authors lean towards 4.0×10^{-9} and ended up reaching a very ancient divergence time for dog domestication. I am not sure this is well supported by the genetic and archaeological data. The authors should be aware of these uncertainties in light of the previous findings from the field.

Estimation of the mutation rate is undeniably controversial across a variety of taxa (not just canids), and we are aware of the various issues that surround this. We feel that the reviewer’s concern on this point arose due to a lack of clarity in the description of our results.

As the reviewer notes, Wang et al 2013 use a rate of 6.6×10^{-9} /bp/gen, which is based upon a 3-year generation time and a per-year mutation rate of 2.2×10^{-9} /bp, derived from phylogenetic comparisons of diverse mammalian taxa (Kumar and Subramanian, PNAS 2002). More recent analysis in humans and other species utilizing direct measurements, ancient DNA comparisons, and other methods have generally supported a slower mutation rate than that implied by phylogenetic comparisons. Indeed, as the reviewer notes, Skoglund et al. argue for a rate of 4×10^{-9} /bp/gen based on comparison with an ancient wolf genome. In this study, we estimate an upper bound value for the fastest possible mutation rate that could be reconciled with the ages of our samples (i.e. it could be any value lower than this, as also mentioned in the estimation of the rate by Skoglund), and find that this upper bound is 5.6×10^{-9} /bp/gen (with a confidence interval on the upper bound, not on the estimated rate itself, of $3.7-7.4 \times 10^{-9}$) (note the slight decrease in these values compared to the original manuscript because of the new f4 ratio analysis (see above)). This would put the phylogenetic rate (6.6×10^{-9}) right at our estimated upper bound. Since this faster rate is at our upper bound and since it is not compatible with the dating of Skoglund et al., we favor using a slower mutation rate for conversion to years. While we present the uncalibrated values so that readers can easily apply their own preferred rate, we believe that our choice is well justified given the current state of knowledge.

We have modified the manuscript text to more clearly state that we have determined the upper bound of mutation rates that are compatible with the age of our samples and to explain our use of the 4×10^{-9} value.

Pg 11 paragraph 1:

“Specifically, we used the age of the HXH sample to set an upper bound for the yearly mutation rate μ , as the sample must be younger than the time in years since divergence of HXH and modern European dogs. Given that the sample is 7,000 years old, we infer that an upper bound

for μ is 5.6×10^{-9} per generation (assuming a 3 year generation time, with a 95% CI for the upper bound of 3.7×10^{-9} to 7.4×10^{-9} , Supplementary Figure S13.5.1.c). This upper bound, which represents the highest mutation rate potentially compatible with the age of our samples, is consistent with the rate of $\mu=4 \times 10^{-9}$ per generation suggested by both Skoglund et al.³² and Frantz et al.¹¹, two rates also calibrated by ancient samples. In contrast, a mutation rate of 6.6×10^{-9} , based on phylogenetic analyses^{33,34}, would be coincident with our estimate of the upper bound and is not compatible with Skoglund et al. or Frantz et al.”

Reviewer #1 (Remarks to the Author):

The revision is quite satisfactory, I have no further comments.